# Impact of Protein Nanoparticle Shape on the Immunogenicity of Antimicrobial Glycoconjugate Vaccines

**DOI:** 10.3390/ijms25073736

**Published:** 2024-03-27

**Authors:** Marta Dolce, Daniela Proietti, Silvia Principato, Fabiola Giusti, Giusy Manuela Adamo, Sara Favaron, Elia Ferri, Immaculada Margarit, Maria Rosaria Romano, Maria Scarselli, Filippo Carboni

**Affiliations:** 1Department of Biotechnology, Chemistry and Pharmacy, University of Siena, 53100 Siena, Italy; 2GSK, 53100 Siena, Italy; 3Department of Chemistry, Materials and Chemical Engineering, Politecnico di Milano, 20133 Milano, Italy

**Keywords:** glycoconjugates, protein nanoparticles, vaccines, antimicrobial vaccines

## Abstract

Protein self-assembling nanoparticles (NPs) can be used as carriers for antigen delivery to increase vaccine immunogenicity. NPs mimic the majority of invading pathogens, inducing a robust adaptive immune response and long-lasting protective immunity. In this context, we investigated the potential of NPs of different sizes and shapes—ring-, rod-like, and spherical particles—as carriers for bacterial oligosaccharides by evaluating in murine models the role of these parameters on the immune response. Oligosaccharides from *Neisseria meningitidis* type W capsular polysaccharide were conjugated to ring-shape or nanotubes of engineered *Pseudomonas aeruginosa* Hemolysin-corregulated protein 1 (Hcp1cc) and to spherical *Helicobacter pylori* ferritin. Glycoconjugated NPs were characterized using advanced technologies such as High-Performance Liquid Chromatography (HPLC), Asymmetric Flow-Field Flow fractionation (AF4), and Transmission electron microscopy (TEM) to verify their correct assembly, dimensions, and glycosylation degrees. Our results showed that spherical ferritin was able to induce the highest immune response in mice against the saccharide antigen compared to the other glycoconjugate NPs, with increased bactericidal activity compared to benchmark MenW-CRM_197_. We conclude that shape is a key attribute over size to be considered for glycoconjugate vaccine development.

## 1. Introduction

Vaccines are one of the most important achievements in preventive medicine because they ensure protection against a broad range of infections by stimulating humoral and cellular immune responses. Carbohydrates, found abundantly on the surface of many pathogens, are attractive candidates for vaccine development due to their interaction with both innate and adaptive immunity [1,2,3]. Many licensed vaccines use bacterial cell-surface antigens conjugated to subunit proteins to trigger antigen-specific protective antibodies and long-lasting immune responses by eliciting T-cell responses and memory B cell activation [4,5,6]. However, glycoconjugate vaccines often require adjuvants and booster injections to maintain protective antibody levels due to their low immunogenicity [7].

The field of glycoconjugate vaccines is continually advancing, improving our understanding of key carbohydrate characteristics that can influence the immunogenicity of glycoconjugates. This includes research on new and innovative techniques for creating saccharide antigens, such as synthetic or chemo-enzymatic methods, which have already yielded promising results [8,9]. Concurrently, there is a growing interest in identifying new carrier proteins that could potentially amplify vaccine efficacy. This is achieved by displaying multiple copies of an antigen, thereby mimicking its natural presentation by the pathogen. A multivalent presentation can be achieved using protein nanoparticles that offer the possibility to chemically or genetically incorporate antigens exposed to their surface, mimicking pathogen dimensions [10,11,12,13]. The elicitation of a more efficient immune response relies on the increase in cross-linking between B-cell receptors and antigens presented with consequently augmented B- and T-cell stimulation and activation [14,15]. Moreover, protein nanoparticles have desirable traits such as biodegradability and biocompatibility, and have demonstrated efficient delivery of antigens to the lymphatic system due to their high uptake by dendritic cells [16].

Protein nanoparticle-based vaccines have already demonstrated their immunological value [17,18] with the first approved vaccine being the Hepatitis B virus (HBV) vaccine. HBsAg, which constitutes the envelope of HBV, self-assembled into 22 nm virus-like particles (VLPs) [19] and resulted in 1000 times more immunogenicity than not-assembled HBsAg [20]. The same strategy was also used for the development of Human Papillomavirus (HPV) and Human Hepatitis E virus (HEV) vaccines [21,22,23].

In addition to their use as antigens, protein nanoparticles can also serve as carriers, addressing key challenges in vaccine design. They can carry various biomolecules, including polysaccharides [24,25], viral [26,27,28,29] and bacterial [30,31,32,33] protein antigens, peptides [34,35,36] and glycopeptides [37], nucleic acids [38] and small molecules [39,40] making them versatile platforms for vaccine development. Protein-based nanocarriers have displayed great potential for coronavirus vaccine research [41,42,43], indicating their relevance in addressing current global health challenges.

Despite the licensing of various VLP-based vaccines and continuous progress in conjugation techniques for bacterial saccharide antigens, few glycoconjugated nanoparticles are currently under development in the pre-clinical stage. Carrier protein nanoparticles have been employed to promote a strong T-cell and long-lasting anti-glycan immune response against *N. meningitidis*, *K. pneumoniae,* and *V. cholerae* [44,45,46], to induce nanomolar affinity of antibodies towards pneumococcal saccharide antigens [45,47] or to develop a single-dose vaccine addressing the medical need for a maternal vaccine against Group B *Streptococcus* (GBS) [25]. Additionally, in recent years, Nano-B5 nanoparticles based on the bacterial pentameric AB-5 toxin and designed through computational methods have been successfully bioconjugated to the O-polysaccharides of *Klebsiella pneumoniae* and *Shigella flexneri* with promising prophylactic effects in mice, proving to be a potential technology for the development of AMR vaccines [48].

Despite these few examples, no glycoconjugated nanoparticle vaccine candidates have advanced to the clinic. Still, limited knowledge is available on the optimal nanoparticle characteristics for glyconanoparticle vaccine candidates. In particular, the ideal size and shape to improve the delivery of saccharide antigen and enhance the immune response have not been investigated. The correlation between nanomaterial characteristics and their immune response has been primarily concentrated on gold and polymers, with minimal attention to protein nanoparticles, especially for glycoconjugate vaccine development [17,49].

In the present study, we investigated how the size and shape of protein nanoparticles could influence the elicitation of an IgG response against bacterial saccharide antigens exposed to their surface. For this purpose, we used self-assembling proteins, manipulated in vitro, to get the desired shape and size, conjugated them to *Neisseria meningitidis* type W saccharide antigens, and tested their immune response in a murine model.

## 2. Results

### 2.1. Production of Glycoconjugated Protein Nanoparticles

For our purposes, we looked for bacterial proteins from the literature that could: (i) cover the most represented particle shapes; (ii) be modulated in terms of size by varying their production methods; and (iii) present surface-exposed lysines as preferential sites for glycoconjugation.

We selected the Hemolysin-coregulated protein (Hcp1) from *Pseudomonas aeruginosa* to generate nanorings and nanotubes, and *Helicobacter pylori* ferritin to produce nanospheres.

Previous research by Ballister et al. reported that Hcp1 can spontaneously assemble into homohexameric rings with an outer diameter of 9.0 nm and a height of 4.4 nm [50,51]. Moreover, the double mutant Gly-90/His and Arg-157/His (Hcp1cc) can stabilize through the formation of disulfide-bonded nanotubes up to approximately 100 nm in height [50].

After *E. coli* production and purification, Hcp1cc immediately originated a heterogeneous population of nanotubes with an average height of 20 nm, resulting from 4 to 5 assembled rings and reaching a maximum of 40 nm from 8 to 9 rings (Figure 1b). Longer nanotubes were generated by incubating Hcp1cc with the reducing agent dithiothreitol (DTT) to break disulfide bonds, followed by protein concentration and extensive dialysis. After 5 days of dialysis, nanotubes up to 60–80 nm in height, composed of 14–18 assembled rings, were observed by negative-stain Transmission Electron Microscopy (TEM) (Figure 1c). In summary, as shown in Figure 2 and Table 1, we produced ferritin nanospheres and Hcp1cc nanorings of comparable dimension (9–10 nm in diameter) and two populations of Hcp1cc nanotubes, one with a maximum height of 40 nm and a second one of 60 nm.

We selected oligosaccharides (OS) derived from *Neisseria meningitidis* type W (MenW) capsular polysaccharide as a model antigen to be chemically conjugated to self-assembling nanoparticles, applying the same conjugation strategy used for the commercial glycoconjugate vaccine MenACWY-CRM_197_ (MENVEO), which represents our reference [52,53].

The Hcp1cc nanotubes were incubated overnight in the presence of the activated MenW OS, and the resulting rod-shaped glycoconjugates were purified from unreacted oligosaccharides.

To obtain glycoconjugated nanorings, Hcp1cc nanotubes were incubated with DTT (Figure 1a). The resulting Hcp1cc hexamers were then incubated with the activated MenW OS, maintaining reducing conditions. The resulting glycoconjugated nanorings were purified from unreacted oligosaccharide, and DTT. Despite the removal of the reducing agent, nanoring association into nanotubes was prevented by the presence of the conjugated saccharides that shielded the formation of disulfide bonds.

Finally, *Helicobacter pylori* ferritin was selected due to its well-known capability of self-assembling into spherical nanoparticles of 10 nm diameter, exploring an additional shape with dimensions comparable to Hcp1 nanorings. Ferritin NPs were produced in *E. coli* and purified via Size Exclusion Chromatography (SEC), while their correct assembly was assessed via TEM analysis (Figure 1d). As for the other nanoparticles, ferritin was conjugated to MenW OS.

### 2.2. Advanced Characterization of Glycoconjugated Nanoparticles

The glycoconjugated nanospheres, nanorings, and nanotubes were widely characterized thanks to a panel of analytical techniques that allowed us to highlight their different features. In particular, glycoconjugation was assessed via Sodium Dodecyl Sulphate-Polyacrylamide Gel Electrophoresis (SDS-PAGE) and Size Exclusion High-Performance Liquid Chromatography (SE-HPLC). The nanoparticle integrity, once glycoconjugated, was confirmed by negative-stain Transmission Electron Microscopy (TEM) analysis. Saccharide and protein contents were quantified, respectively, via high-performance anion-exchange chromatography with pulsed amperometric detection (HPAEC-PAD) and Micro Bicinchoninic Acid (mBCA), a colorimetric assay. Finally, the asymmetric flow field flow fractionation (AF4) technique was mainly utilized to assess sample heterogeneity and to cross-check the saccharide/protein molar ratio using an orthogonal tool.

Hcp1cc nanostructures, both rings and tubes, were reduced to the monomeric form due to the break of disulfide bonds in the presence of DTT and detected as a single band at 18 kDa in SDS-PAGE experiments (see lines 1B and 3B in Figure 3). Instead, in non-reducing conditions, nanotubes were described by multiple bands (4–5 bands) representing the number of rings composing the nanoparticle (line 3A in Figure 3). In the presence of SDS, rings dissociated while the disulfide bonds among the monomers that compose the height remained intact, providing information about the number of rings. After the oligomerization process, a major number of bands could be detected with increased intensity, especially for bands at high MWs because longer structures became more represented (see line 5 in Figure 3).

After glycoconjugation, SDS-PAGE experiments showed a high MW smear (see lines 2A, 4A, and 6A in Figure 3), confirming successful conjugation.

Initial SE-HPLC experiments produced Hcp1cc profiles difficult to interpret, probably due to the high MW dimensions and a possible interaction with the stationary phase of the column, which was eliminated by adding 0.1% sodium dodecyl sulfate to the elution buffer. An increase in size was observed moving from nanorings to longer nanotubes, with an increase in the heterogeneity of the sample and the shift of the peak towards shorter retention times corresponding to the increase in MW (black profiles in SE-HPLC in Figure 3). After glycoconjugation, an additional shift of peaks at lower retention times in SE-HPLC confirmed the conjugation (red profiles in SE-HPLC in Figure 3).

TEM analysis confirmed the integrity of Hcp1cc particles, showing that glycoconjugation did not impact the structure assembly once formed (Figure 4b,c). Moreover, the MenW-nanoring sample did not show nanotube structures by TEM but a ring shape with a homogeneous diameter size of 9 nm, confirming the inhibition of ring–ring association caused by the presence of the oligosaccharides (Figure 4a).

A high glycosylation degree was also confirmed by calculating the saccharide/protein ratio (Table 2; Section 4.13). Nanorings were conjugated with an average of 2 MenW chains per monomer, obtaining an average of 12 antigen copies per ring. MenW-Hcp1cc nanotubes resulted in even more glycosylation, displaying an average of 18–24 chains of MenW OS per ring repeated up to 14 times for the longest nanotubes.

In *H. pylori* ferritin glycoconjugates, all 24 monomers that compose the spheric nanoparticle were derivatized with the MenW OS. In fact, analyzing the glycoconjugates via SDS-PAGE, the ferritin monomer at 19 kDa was poorly detected and turned into a high MW smear (see line 2A in Figure 3). Moreover, the efficiency of the conjugation was also demonstrated by the SE-HPLC profile, which showed a single peak with a lower retention time compared to the starting nanoparticles and the absence of detectable unconjugated protein (see SE-HPLC profile in Figure 3). The high degree of glycoconjugation did not appear to impact the nanoparticle structure, which remained spherical with a peculiar diameter of 10 nm as observed in TEM analysis (Figure 4d). From saccharide quantification, almost 2 chains of MenW OS were conjugated to each monomer of ferritin nanoparticle, obtaining an average of 45 copies of antigen exposed per particle (Table 2).

The reference MenW-CRM_197_ was also prepared and characterized. Differently from the multiple copies that were displayed on the nanoparticles under investigation, a final glycosylation degree (*w*/*w*) of 0.7 was determined, which resulted in an average of 5 chains of MenW oligosaccharides exposed to the protein.

To overcome the limitations of traditional chromatography, AF4 analysis was set up. This technique separates large aggregates and particles (e.g., LNPs, VLPs, and polymers) based on their size [54,55]. This separation is obtained by the difference in mobility in the flow field induced by the liquid flow over an inert membrane and across the channel. The presence of different detectors in line, such as multi-angle light scattering (MALS), UV-Vis absorbance at 280 nm, and differential refractive index (dRI), allows the determination of population distribution in terms of molar mass and saccharide content using a protein conjugation tool.

Analyzing MenW-Hcp1cc nanoparticles, an increase in MW was observed along with an increase in the oligomerization state of the Hcp1cc protein. In particular, one single main population of MenW-nanotubes was detected, while two main distinct populations at higher MW were detected for MenW-Hcp1cc nanotubes that further increased after the oligomerization process as expected. The presence of a single and defined population on MenW-Ferritin was also confirmed by AF4 analysis, where a sharp and single peak is described by HPLC analysis (Figure 3).

The saccharide/protein molar ratio can be calculated using AF4 analysis by integrating UV and RI signals to calculate the protein and saccharide contributions in terms of MW composing the glycoconjugate. The ratio calculated for each glycoconjugate was consistent with the ones calculated using the results obtained by the HPAEC-PAD and mBCA assays (Table 2).

### 2.3. Evaluation of Immune Responses Elicited by Glycoconjugated Nanoparticles in the Murine Model

Groups of 10 CD-1 mice were immunized with three doses of MenW-Hcp1cc or MenW-Ferritin glycoconjugates (1 µg of MenW/dose), administered three weeks apart, in comparison with the benchmark MenW-CRM_197_ [56,57]. Each antigen was adjuvanted with AS01, a liposome-based adjuvant that contains two immunostimulants, a TLR4 ligand, 3-*O*-desacyl-4′-monophosphoryl lipid A (MPL), and a saponin, QS-21.

Two weeks after the third dose, sera were collected to measure IgG titers by ELISA assay, using MenW capsular polysaccharide as a coating reagent. After three doses, MenW-Ferritin elicited a statistically significant higher immune response (*p* values < 0.0001) compared to both MenW-Hcp1cc nanorings and nanotube conjugates, in which some mice with no or very low IgG titers were detected (Figure 5a).

Sera of immunized mice were also tested for functional activity in the serum bactericidal assay using human serum as a source of complement (hSBA), assessing the protective response to meningococcal vaccination. Despite the variability in the response of individual mice, MenW-Ferritin was able to elicit a higher bactericidal activity compared to MenW-Hcp1cc nanorings (statistically significant, *p* value < 0.033) and to the benchmark vaccine (Figure 5b). Thus, MenW-Ferritin performed best among the tested vaccine candidates.

## 3. Discussion

Despite the fact that nanoparticles have been widely examined in the field of vaccines, their role as carriers in glycoconjugate vaccine development has not been equally explored. This lack limits our understanding of the key characteristics that a glycoconjugated nanoparticle should have to induce an effective and potent immune response. Our study aimed to investigate how attributes such as size and shape of the NPs influence the immune response to saccharide antigens exposed to their surface. The ultimate goal is to guide the future design and selection of the best nanoparticles for the development of promising glycoconjugate candidates against the bacterial target.

In the present work, we identified from the literature the self-assembling Hcp1cc protein, which spontaneously arranges into ring-shaped hexamers with a diameter of 9 nm [50,51]. These nanoparticles can oligomerize into nanotubes with a height of 40 nm that can be extended up to 80 nm via in vitro manipulation. This property allowed us to study the effects of different geometry and the impact of carrier size on antigen immunogenicity. In addition, we selected a second self-assembling protein, *H. pylori* ferritin [27,58,59,60,61], which forms spherical nanoparticles with comparable diameter to examine the impact of shapes while maintaining similar sizes.

All selected nanoparticles were successfully chemically conjugated to activated MenW oligosaccharides thanks to the presence of multiple lysines on their surface, achieving a high glycosylation degree. Several parameters of the nanoparticles before and after conjugation were controlled using a panel of complementary techniques, including AF4, a gold standard worldwide for nanoparticle characterization [54,55,62,63,64,65].

Finally, glycoconjugated nanoparticles were administered in three AS01-adjuvanted doses in mice using MenW-CRM_197_ as a benchmark vaccine. MenW-Ferritin was the most immunogenic among the tested MenW-Nanoparticles, stimulating a strong antibody response in vivo against the MenW oligosaccharide. Interestingly, the increase in nanotube size between 10 and 100 nm did not affect the immune response. Most importantly, MenW conjugation to spherical ferritin nanoparticles resulted in the highest bactericidal activity in a functional assay among all tested samples.

Our data outline the major role played by the nanoparticle shape on antigen immunogenicity, with spherical nanoparticles and eliciting the highest protective immune response in vivo compared to elongated nanotubes. Our findings extend to the protein-based nanoparticles and the conclusions of recent observations reported on the role of the size and shape of gold and polymeric NPs. Niikura et al. demonstrated that spheric gold nanoparticles (AuNPs) coated with West Nile virus envelope protein (WNVE) induced 50% more WNVE-specific IgG antibodies in mice compared to rods [66]. Also, Toraskar et al. administered in vivo AuNPs coated with tripodal Tn-glycopeptide antigen, showing that the small, spherical-shaped AuNPs induced an effective anti-Tn-glycopeptide IgG response compared with rod-shaped AuNPs [67]. Kumar et al. also concluded that among different particle types, spherical polystyrene ovalbumin-conjugated particles generated in vivo stronger immune responses by inducing the most potent Th1 and CD8+ T-cells [68]. Several mechanisms have been proposed to explain the superiority of spherical nanoparticles, including more efficient cellular uptake [69,70], better APC activation, and faster trafficking to lymph nodes [71,72].

Many comparative studies have demonstrated the influence of particle shape on DC activation, observing that inorganic spherical nanoparticles are more potent in stimulating DCs compared to their non-spherical counterparts, relying on upregulation of CD83 and CD86, a more efficient internalization [69,70] and cell endocytosis for spherical over rod-shaped nanoparticles [73,74].

It has been hypothesized that elongated particles may be less efficiently taken up because of the reduced contact area when attached to the cells along their minor axis, whereas spherical nanoparticles always expose an optimal contact area to the cell membrane, resulting in less energy cost internalization [75,76,77,78].

Our data are also in line with evidence on inorganic nanoparticles, indicating that an increase in size can be beneficial for cellular uptake [79,80]. Indeed, gold nanoparticles of 20–200 nm efficiently enter the lymphatic system and reach the lymphatic organs directly within hours of injection [71,72]. On the contrary, particles larger than 200 nm do not efficiently enter lymph capillaries in a free form but need DCs transportation requiring 24 h [72,81]. Furthermore, DC preferentially take up 40 nm nanoparticles over 200 nm [82]. Moreover, smaller particles are most potent in inducing IFN-γ-mediated Th1 immunity compared to particles in the 100 nm range, which induce stronger IL-4 responses and Th2 responses. It was suggested that smaller (<100 nm) particles may enter APCs through one of the mechanisms used by viruses, such as clathrin-coated pit-mediated uptake, which may induce a stronger Th1 immune response [83,84].

Moreover, it is crucial to highlight that the nanoparticles being compared, although they share a similar molecular weight as the monomeric unit, contain distinct T-cell epitopes. This difference could potentially influence the immune responses they trigger, a topic that warrants further exploration.

In conclusion, our study highlights the potential of self-assembling protein nanoparticles for glycoconjugate vaccine development and remarks that shape over size is a key factor in enhancing the efficiency of the immune response. We present the first example of glycoconjugated *H. pylori* ferritin as an efficient carrier protein for efficient delivery of bacterial saccharide antigen. We hypothesize that, compared to inorganic nanoparticles, the ferritin scaffold could offer a repertoire of effective peptidic T-cell epitopes to the immune system, thereby enhancing the protective response.

Overall, our findings outline that protein nanoparticles may represent a promising strategy to develop novel glycoconjugate vaccines against bacterial pathogens.

## 4. Materials and Methods

### 4.1. Protein Nanoparticles Expression in E. coli

The mutated Hcp1cc (G90C + R157C) was expressed as a His6-tagged protein in *Escherichia coli* BL21(DE3). First, a pre-culture in LB (Tryptone 10 g/L, yeast extract 5 g/L, NaCl 10 g/L) medium supplemented with Ampicilin 100 mg/L from glycerol stock of BL21(DE2) pET303/CT-HIS-HCP1 was incubated at 37 °C and 190 rpm for 16 h. The pre-culture was used to set up batch cultivation of protein in 2YT medium (16 g/L Trypton, 10 g/L Yeast extract, 5 g/L NaCl pH 7.6) supplemented with Ampicillin 100 mg/L, under shaking at 37  °C for 2 h, until OD value of 0.5, followed by induction with 0.5 mM IPTG for 5 h at 37 °C at 180 rpm.

*H. pylori* Ferritin was expressed as a tag-less protein in *Escherichia coli* BL21(DE3). First, a pre-culture in LB (Tryptone 10 g/L, yeast extract 5 g/L, NaCl 10 g/L) medium supplemented with Kanamicin 50 mg/L from colonies grown on selective plates of BL21(DE2) was incubated at 37 °C, 180 rpm for 7 h. The pre-culture was used to set up batch cultivation of protein in LB medium supplemented with Kanamicin 50 mg/L, under shaking at 32 °C for 10 h, followed by induction with 1 mM IPTG for 16 h at 25 °C at 160 rpm.

### 4.2. Protein Nanoparticles Purification

Crude cell lysate obtained after His6-tagged Hcp1cc expression in *E. coli* was purified via Co^2+^ affinity chromatography using a His-Trap TALON fast flow crude (Cytiva, Marlborough, MA, USA) column. The column was connected to a peristaltic pump of the instrument ÄKTA pure™ (Cytiva) and washed with distilled water and then equilibrated with buffer 50 mM NaPi 300 mM NaCl pH 8. Crude cell lysate was applied to the column, and the flow through was discarded. The column was first washed with 20 CV of buffer 50 mM NaPi and 300 mM NaCl, then a gradient of buffer 50 mM NaPi, 300 mM NaCl, and 500 mM imidazole in buffer 50 mM NaPi, 300 mM NaCl was applied to elute the target protein. Hcp1cc protein was eluted in buffer 50 mM NaPi, 300 mM NaCl, and 500 mM imidazole after IMAC purification. The buffer was exchanged into buffer 500 mM NaCl, 50 mM TRIS pH 7.5 via tangential flow filtration using a hydrosart membrane with a cut-off of 30 kDa (Sartoriuos Stedim biotech, Aubagne, France).

Crude cell lysate obtained after tag-less *H. pylori* Ferritin expression in *E.coli* was purified first in a Sephacryl S-300 HR 26/60 column (Cytiva) and eluted in buffer PBS 1x. Then, a second purification was performed in the HiLoad Desalting 26/10 (Cytiva) column, and the target protein was eluted pure in PBS 1x.

### 4.3. Oligomerization Process—In Vitro Reassembly for the Generation of Long Hcp1cc Nanotubes

Purified Hcp1cc nanotubes were treated with 5 mM DTT to fully reduce nanotubes into ring form and concentrated at 40 mg/mL in 500 mM NaCl, 50 mM TRIS pH 7.5, 10% glycerol, and 5 mM DTT. Protein was dialyzed using a dialysis tube with a membrane 2K cut-off (MINI Dialysis Unit Thermo Scientific Slide-A-Lyzer^®^, Waltham, MA, USA) against a buffer containing 5% PEG 3350, 50 mM trisodium citrate, 100 mM HEPES pH 7.5, and 2 mM β-mercaptoethanol until a smooth paste formed (after 5 days of dialysis). This paste was diluted to 10 mg/mL of protein in 500 mM NaCl, 50 mM TRIS pH 7.5, and 10% glycerol. The Hcp1cc protein solution obtained was analyzed first in TEM and SDS-PAGE to verify the presence of long nanotubes.

### 4.4. MenW Oligosaccharide Sizing

Oligosaccharides from *Neisseria meningitidis* type W (MenW) capsular polysaccharides have been selected as model antigens to be chemically conjugated to self-assembling nanoparticles. In particular, applying the same conjugation strategy used for the commercial vaccine MenACWY-CRM (MENVEO) [52,53], the MenW capsular polysaccharide has been sized by controlled acid hydrolysis to obtain MenW oligosaccharides with a defined length of 7.8 kDa (average) and subsequently activated via reductive amination and coupling to bis-N-hydroxysuccinimidyl adipate (SIDEA) linkers generating half-ester residues prompting to react with amine groups present on protein nanoparticles in the conditions reported below [52,56,57].

### 4.5. Nanoparticles Conjugation to MenW Oligosaccharide

To produce MenW-Hcp1cc nanorings, Hcp1cc nanotubes at 40 mg/mL in 500 mM NaCl, 50 mM TRIS pH 7.5, and 10% glycerol were disassembled in the presence of 5 mM DTT. The resulting Hcp1cc nanorings were incubated with 10 equivalents of SIDEA activated MenW oligosaccharide over night at room temperature under gentle shaking. The unconjugated oligosaccharide was extensively removed by performing serial centrifugal filtration (30 kDa) with reducing buffers of 500 mM NaCl, 50 mM TRIS pH 7.5, and 10% glycerol with 5 mM DTT to maintain Hcp1cc in ring form. Finally, the purified MenW-Hcp1cc nanorings were recovered in 500 mM NaCl, 50 mM TRIS pH 7.5, and 10% glycerol.

Hcp1cc nanotubes and longer ones obtained after the oligomerization process were incubated at around 10 mg/mL in 500 mM NaCl, 50 mM TRIS pH 7.5, 10% glycerol, and 30 equivalents of MenW oligosaccharide for one night at room temperature under gentle shaking. The resulted glycoconjugated nanotubes were purified from unreacted MenW oligosaccharide by performing serial centrifugal filtration (30 kDa). The purified MenW-Hcp1cc nanotubes and MenW-Hcp1cc long nanotubes were both recovered in 500 mM NaCl, 50 mM TRIS pH 7.5, and 10% glycerol.

*H. pylori* Ferritin at 10 mg/mL in PBS 1x was incubated with 10 equivalents of SIDEA activated MenW oligosaccharide over night at room temperature under gently shaking. The glycoconjugates obtained were purified from unreacted oligosaccharide performing serial centrifugal filtration (100 kDa). The purified MenW-Ferritin was recovered in PBS 1x.

### 4.6. TEM

A volume of 5 µL of samples, diluted in PBS 1x at 20 ng/microliter, were loaded for 30 s onto glow-discharged copper 200 or 300-square mesh grids. Blotted the excess, the grid was negatively stained using NanoW for 30 s, and let air dry. The samples were analyzed at UNISI using a Tecnai G2 spirit, and the images were acquired using a Tvips TemCam-F216 (EM-Menu 4 software).

### 4.7. SDS-PAGE

Sodium Dodecyl Sulfate-Polyacrilamide Gel Electrophoresis (SDS-PAGE) was performed on 4–12% pre-casted polyacrylamide gel (NuPAGE^®^Invitrogen, Carlsbad, CA, USA) using MOPS 1x as running buffer (NuPAGE^®^Invitrogen). A total of 5 or 10 μg of protein nanoparticles or glycoconjugated nanoparticles were mixed with 3 μL of 4x LDS, 3 μL dithiothreitol (DTT) at 0.5 M when needed, and boiled at 90 °C for 1 min. Samples and prestained protein molecular markers (Pecision Plus Protein Standard Dual color-Biorad) were loaded on precast polyacrylamide gels and run at 200 V and 180 mA for 45 min. Gels were stained with Coomassie ProBlue Safe stain (Giotto, Padua, Italy) for protein visualization.

### 4.8. SE-HPLC

Hcp1cc nanoparticles and corresponding MenW-Hcp1cc nanoparticles were analyzed in a Superdex 200 10/300 GL (Cytiva) column in isocratic elution with 50 mM TRIS, 500 mM NaCl pH 7.5, and 0.1% SDS as running buffer, performing 40 min of run at 1 mL/min of flow rate using an Ultimate 3000 HPLC system (Thermo Scientific) equipped with a photodiode array detector and a multiple-wavelength fluorescence detector. While *H. pylori* Ferritin and MenW-Ferritin were analyzed in a TSK4000PW column (7.8 × 300 mm, Tosoh, Tokyo, Japan) in isocratic elution with 100 mM NaPi, and 100 mM Na_2_SO_4_ at pH 7.2 as running buffer, performing 50 min of run at 0.5 mL/min of flow rate using the HPLC system reported above. The resulting chromatographic data were integrated and processed using Chromeleon™ 7.2 software.

### 4.9. AF4

The asymmetric flow field-flow fractionation (AF4) system (AF2000-Postnova, Landsberg am Lech, Germany) consisted of an isocratic LC pump (PN1130, Postnova analytics, Salt Lake City, UT, USA), autosampler (PN5300 series, Postnova analytics), column oven (PN4020, Postnova analytics), MALS detector (PN3621, Postnova analytics), UV-Vis detector (SPD-20A prominence, Postnova analytics), and refractive index (RI) instrument (PN3150, Postnova analytics). A microchannel with a 350 μm spacer and a 10 kDa regenerated cellulose membrane was used for all separations.

10 μL of sample are injected in the separation channel at 0.20 mL/min and subjected to a cross flow at 1 mL/min and a focus pump at 1.30 mL/min for 3 min. Elution step, in which focus flow decreases to 0 mL/min and the sample is able to elute along the channel for 50 min, subjected to cross flow that decreases in 40 min to 0.1 mL/min in a parabolic manner. PBS 1x or 0.9% NaCl filtered 0.1 µm was used as the mobile phase, and 10 µL of sample was injected. The UV and MALS signals were integrated to obtain population distribution analysis using the Zim plot method and parameters extinction factor of 1.313 [mL/(mg×cm)] for Hcp1cc and 1.056 [mL/(mg×cm)] for ferritin, calculated from protein sequences, and dn/dc of 0.185 provided by Postnova and literature [85,86,87].

### 4.10. AF4 Protein Conjugates Tool for Saccharide/Protein Molar Ratio

For each glycoconjugated nanoparticle, a UV signal was integrated to calculate protein contribution in terms of MW using the parameters dn/dc: 0.185 mL/g and an extinction factor of 1.313 [mL/(mg×cm)] calculated for Hcp1cc and 1.056 [mL/(mg×cm)] for ferritin, while saccharide contribution in terms of MW was calculated from the dRI signal using dn/dc: 0.147 mL/g [85,86,87]. The saccharide and proten MW contributions obtained were divided, respectively, for saccharide MW of 7.8 kDa and protein MW of 18.3 kDa for Hcp1cc and 19.3 kDa for ferritin, and compared to obtain the final molar ratio.

### 4.11. Quantification of Protein Content by Colorimetric Assay

Protein concentration was determined by the Pierce™ Micro BCA Protein Assay Kit (mBCA kit, Thermo Fisher Scientific) according to the manufacturer’s instructions and using the provided Pierce™ Bovine Serum Albumin (BSA, Thermo Fisher Scientific) as standard. Each sample was diluted in duplicate, preparing a 5-point calibration curve using a BSA starting solution of 20 µg/mL. Then, the three reagents of the mBCA kit are properly mixed following the manufacturer’s instructions and added to the sample, which is heated at 60 °C in a bath of water for 1 h. The samples are finally transferred to a plastic disposable cuvette and read at the spectrophotometer at 562 nm (Evolution 260 Bio Spectrophotometer, Thermo Scientific, Thermo INSIGHT software 2.1.175). The final protein concentration (µg/mL) is quantified based on the BSA calibration curve.

### 4.12. Quantification of MenW Oligosaccharide Content by HPAEC-PAD

Glycoconjugated nanoparticles were treated at trifluoroacetic acid (TFA) with a final concentration of 2 M. Samples were heated at 100 °C for 2 h, then refrigerated at 4 °C for 15 min and finally dried on a Speed Vac at room temperature overnight. Samples to be analysed were first dissolved in ultrapure water and then filtered (0.45 µm). Analysis of the hydrolysed products was performed using a Dionex ICS5000 (Thermo Fisher Scientific), equipped with a CarboPac PA1™ column and a CarboPac PA1™ guard. Separation was performed with a flow rate of 1 mL/min using isocratic elution of NaOH 15 mM for 20 min, followed by a regeneration step with NaOAc 50 mM/NaOH 100 mM for 10 min, and reconditioning in the starting condition for 15 min. The effluent was monitored using an electrochemical detector in pulsed amperometric mode with a gold working electrode and an Ag/AgCl reference electrode. The resulting chromatographic data were integrated and processed using Chromeleon™ 7.2 software. Galactose concentration was determined using calibration curves set up with commercial stardard galactose and then converted to MenW oligosaccharide concentration (µg/mL) by using a conversion factor that takes into account the weight of galactose in the repeating unit structure.

### 4.13. In Vivo Experiment

Animal treatments were performed in compliance with Italian legislation (Dig 26/2014), EU Directive 63/2010, and GSK Animal Welfare Policy and Standards, and approved by the institutional review board (Animal Ethical Committee) of GSK Vaccines Siena, Italy.

Groups of 10 CD-1 female mice, kept in an AAALAC-accredited facility, were immunized intramuscularly (IM) with MenW glycoconjugate nanoparticles produced (1 μg of MenW/dose) adjuvanted with 2.5 μg/dose of AS01, a liposome-based adjuvant that contains two immunostimulants: a TLR4 ligand, 3-*O*-desacyl-4′-monophosphoryl lipid A (MPL), and a saponin, QS-21. MenW-CRM_197,_ adjuvanted by AS01, was used as a control. Three different immunizations were performed on days 1, 22, and 43, collecting sera at day 57.

### 4.14. ELISA Analysis

The antibody response induced by the glycoconjugates against the polysaccharide has been measured by ELISA. Microtiter plates (96 wells, Thermo Scientific) have been coated with the MenW polysaccharide by adding 100 μL/well of a 5 μg/mL polysaccharide solution in PBS 1x at pH 8.2, followed by incubation overnight (o.n.) at 4 °C. After washing three times with PBS 1x with 0.05% of Tween 20 (Sigma, Saint Louis, MO, USA) (tPBS). A blocking step has been performed by adding 100 μL of BSA solution at 3% in tPBS and incubating the plates for 1 h at 37 °C, then aspirating to remove the solution. Two-fold serial dilutions of test and standard sera in tPBS were added to each well. Plates were then incubated at 37 °C for 2 h, washed with tPBS, and incubated for 1 additional hour at 37 °C with either anti-mouse IgG-alkaline phosphatase whole molecule (Sigma-Aldrich, Saint Louis, MO, USA) diluted 1:2000 in tPBS. After washing, the plates were incubated with 100 μL/well of 1 mg/mL p-Nitrophenyl Phosphate (pNPP sodium salt hexahydrate tablet—Sigma Life Science, Darmstadt, Germany) in 0.5 M of di-ethanolammine buffer pH 9.6. The absorbance was measured using a SPECTRAmax340PC plate reader (Molecular Devices, San Jose, CA, USA) with a wavelength set at 405 nm. IgG concentrations were expressed as relative ELISA Units/mL (EU/mL) and were calculated as the reciprocal of the sera dilution corresponding to OD = 1. Each immunization group has been represented as the geometrical mean (GMT) of the single mouse titers. The statistical and graphical analysis has been performed by GraphPad 5.0 software.

### 4.15. Human SBA

Functional antibodies were measured by a human Serum Bactericidal Activity assay (hSBA) on meningococcal strains 240,070 (reference for meningococcal serotype W) using human serum as a complement source. The hSBA was performed on sera from mice immunized with three doses of AS01 adjuvanted MenW-Hcp1cc nanorings, MenW-Ferritin, and MenW-CRM_197_ and collected 2 weeks after the third immunization.

Bacteria were plated on a round agar plate and incubated for 16 h at 37 °C with 5% CO_2_ in a humid atmosphere. The day after, single colonies were inoculated into Mueller–Hilton Broth (MHB) containing 0.25% *w/v* glucose and incubated at 37 °C at 150 rpm until the culture reached OD = 0.24–0.26. After that, bacteria were diluted 1:20,000 or 1:30,000 in SBA buffer (Dulbecco’s saline phosphate buffer with 0.1% glucose and 1% BSA (Bovine Serum Albumin) supplemented with heparin solution 5 U/mL and salt solution (MgCl_2_ 0.01 M, CaCl_2_ 1.5 mM).

The assay was assembled in a sterile 96-well flat-bottom microplate in a final volume of 40 µL/well, where mouse sera were serially diluted (ten 2-fold dilution steps) in SBA buffer and added to each test sample, which was incubated with bacteria and human complement at 37 °C with 5% CO_2_ for 1 h. Then, 150 μL of agar (TSB 0.7% agar) were added, and the plate was incubated overnight at 37 °C with 5% CO_2_. Bacteria with active human complement (AC) in the absence of a serum sample are the control used to exclude complement toxicity and determine 100% bacterial growth. The SBA titer was determined for each test sample as the reciprocal of the sample dilution, giving a killing ≥ 50% with respect to the average number of CFU calculated (using ScanLab software Icarus 1.2.3 ) on the control.

## Figures and Tables

**Figure 1 ijms-25-03736-f001:**
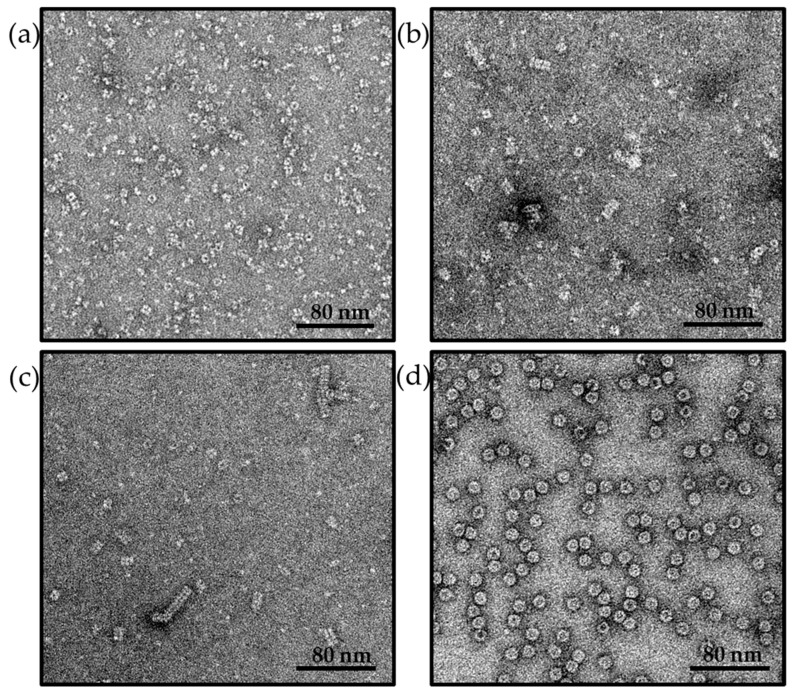
Transmission Electron Microscopy (TEM) of nanoparticles at different sizes and shapes after *E.coli* expression and purification. (**a**) Hcp1cc nanorings produced when Hcp1cc nanotubes are incubated in presence of reducing agent DTT; (**b**) Hcp1cc nanotubes; (**c**) Hcp1cc long nanotubes produced after oligomerization process; (**d**) *H. pylori* ferritin.

**Figure 2 ijms-25-03736-f002:**
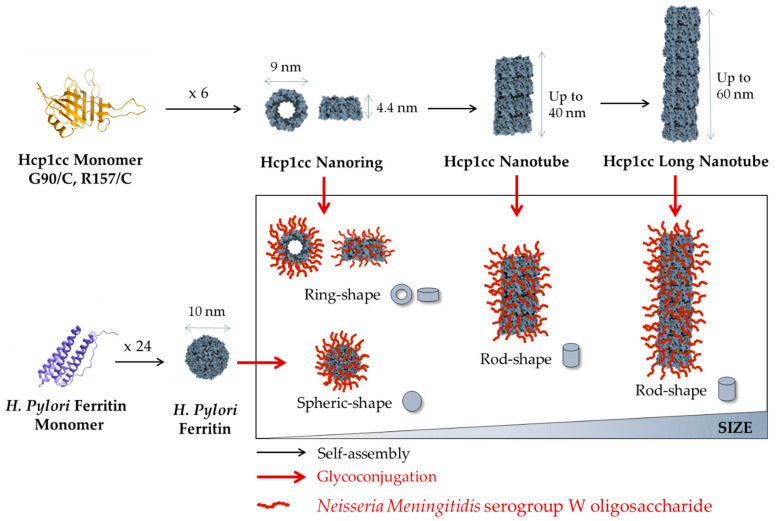
Workflow to produce glycoconjugate nanoparticles and their correlation in terms of sizes and shapes.

**Figure 3 ijms-25-03736-f003:**
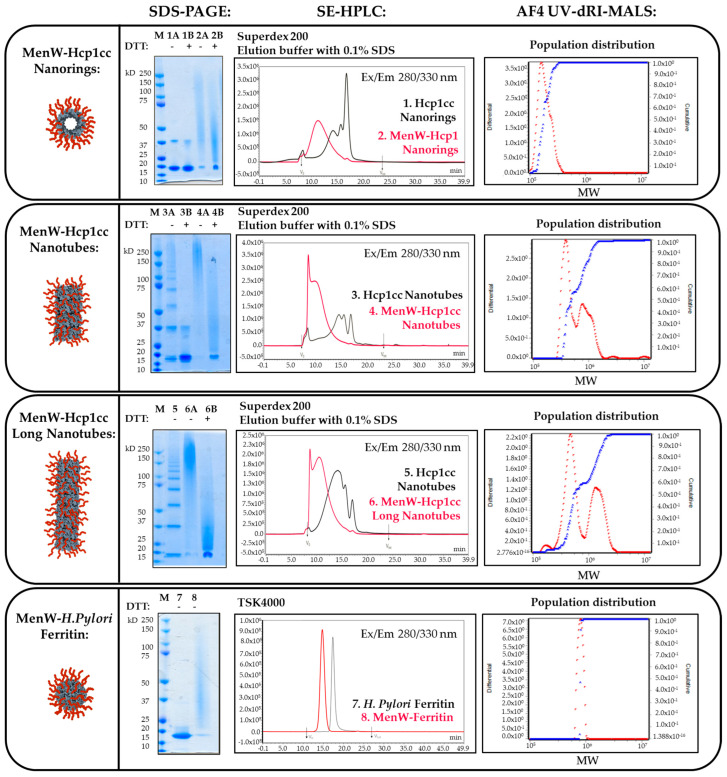
A comprehensive panel of analytical characterization data for glycoconjugate nanoparticles. By order of appearance: SDS-PAGE of nanoparticles and corresponding glycoconjugates in presence and absence of reducing agent DTT; SE-HPLC profiles of nanoparticles and corresponding glycoconjugates; population distribution analysis of glycoconjugates analyzed via AF4 in differential (red line) and cumulative (blue line) distribution.

**Figure 4 ijms-25-03736-f004:**
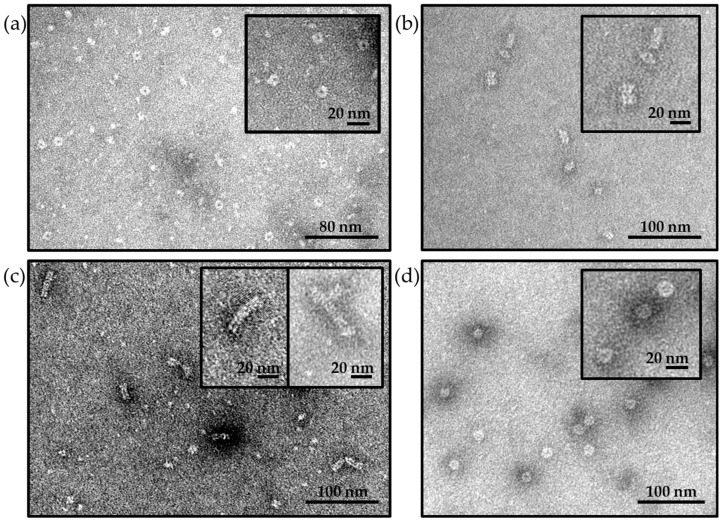
Transmission Electron Microscopy (TEM) of glycoconjugated nanoparticles. The integrity of glycoconjugated nanoparticles and their different sizes and shapes were assessed for (**a**) MenW-Hcp1cc nanorings characterized by only ring structures with a diameter of 9 nm; (**b**) MenW-Hcp1cc nanotubes characterized by nanotubes with a height of 20–40 nm; (**c**) MenW-Hcp1cc long nanotubes characterized by heterogeneous population of rod-shaped nanoparticles at highest order structures up to 60 nm of height; (**d**) MenW-Ferritin characterized by spheric-shaped nanoparticles with a diameter of 10 nm.

**Figure 5 ijms-25-03736-f005:**
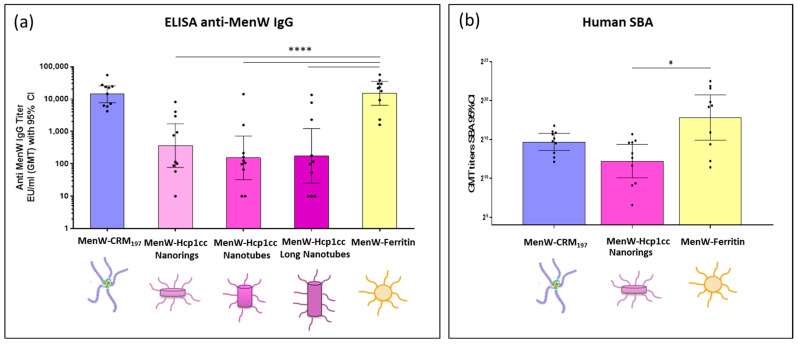
Anti-MenW IgG and Human SBA titers in mice immunized with MenW-nanoparticles or MenW-CRM. (**a**) Immune response in 10 mice per group receiving three AS01-adjuvanted doses of MenW-nanopartilces compared to MenW-CRM_197_ as benchmark vaccine. The *y*-axis indicates the geometric mean IgG titer (RLU/mL), individual mice are indicated by the dots, and the 95% Confidence Interval is indicated by the whiskers. For non-responder sera were assigned GMT titers for half of the LLOQ (10 RU/mL). Mann Whitney test was used to compare benchmark titers with the other immunizations; **** *p*-value < 0.0001 (**b**) Human SBA titers from single mice sera immunized with MenW-CRM_197_, MenW-Hcp1 nanorings, and MenW-Ferritin are reported and compared using Prism. Kruskal-Wallis test; * *p*-value < 0.033. Geometric mean titers and the 95% Confidence Interval is indicated by the whiskers.

**Table 1 ijms-25-03736-t001:** Summary of protein-based nanoparticles used in this study.

	Mw Monomer (kD)	Number of Subunit	Mw Nanoparticle (kD)	Dimension (nm)
Hcp1cc Nanoring	18.3	6 monomers (ring)	109.8	Diameter: 9 Height: 4.4
Hcp1cc Nanotube	18.3	up to 8–9 rings	Up to 878–988	Diameter: 9 Height: up to 40
Hcp1cc Long Nanotube	18.3	up to 14–18 rings	Up to 1537–1976	Diameter: 9 Height: up to 60–80
Ferritin	19.3	24 monomers	463	Diameter: 10

**Table 2 ijms-25-03736-t002:** Saccharide/Protein Ratio.

Glycoconjugate Nanoparticles	HPAEC-PAD/BCA Colorimetric Assay Saccharide/Protein Ratio		AF4 Protein Conjugates Tools Saccharide/Protein Ratio
	*w*/*w*	mol/mol	mol/mol
MenW-Hcp1cc Nanorings	0.9	2.0	2.2
MenW-Hcp1cc Nanotubes	1.3	3.1	4.1
MenW-Hcp1cc Long Nanotubes	1.8	4.2	4.6
MenW-*H. pylori* Ferritin Spheric Nanoparticle	0.8	1.9	1.8
MenW-CRM_197_	0.7	5	Not done

## Data Availability

Data are contained within the article.

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
