# Peer review of "Impact of Protein Nanoparticle Shape on the Immunogenicity of Antimicrobial Glycoconjugate Vaccines"

_ijms, 2024, doi:10.3390/ijms25073736_

Round 1

Reviewer 1 Report

Comments and Suggestions for Authors

The manuscript of Dolce et al. analyze the bactericidal and immunogenic properties of NmW glycoconjugates using protein nanoparticles as carrier proteins. This represents a novel advance as this is an unexplored class of protein carrier. The manuscript is well-written and provides convincing data for production of the protein nanoparticles as well as analytical characterization of the glycoconjugates.  The authors present an exciting potential new avenue for spherical ferritin-based protein nanoparticles as carriers. 

Minor suggestions:

- Language from the figure legend for Figure 3 was somehow incorporated into the manuscript rather than the figure legend text (Lines 209-212) which led to initial confusion.

- While MenW-CRM197 was used as a benchmark, there were no similar characteristics provided (saccharide/protein ratio; saccharide loading) for this glycoconjugate. The manuscript could be improved with some discussion of comparison of these attributes to the ferritin-based glycoconjugate and how the relationship between shape and saccharide loading might further be explored.  

Comments on the Quality of English Language

The manuscript was well-written with only minor editing needed.

Author Response

  • Reviewer 1:

Comments:

The manuscript of Dolce et al. analyzes the bactericidal and immunogenic properties of NmW glycoconjugates using protein nanoparticles as carrier proteins. This represents a novel advance as this is an unexplored class of protein carrier. The manuscript is well-written and provides convincing data for production of the protein nanoparticles as well as analytical characterization of the glycoconjugates.  The authors present an exciting potential new avenue for spherical ferritin-based protein nanoparticles as carriers. 

Minor suggestions:

  1. Language from the figure legend for Figure 3 was somehow incorporated into the manuscript rather than the figure legend text (Lines 209-212) which led to initial confusion.

As per the request, the figure legend for Figure 3 has been reformatted, eliminating the space between the initial and subsequent sentence

  1. While MenW-CRM197 was used as a benchmark, there were no similar characteristics provided (saccharide/protein ratio; saccharide loading) for this glycoconjugate. The manuscript could be improved with some discussion of comparison of these attributes to the ferritin-based glycoconjugate and how the relationship between shape and saccharide loading might further be explored.  

In section 2.2, a concise discussion regarding the characteristics of MenW-CRM197, particularly in terms of glycosylation, has been added. This discussion also includes a comparison with other glyconanoparticles. The new sentence has been incorporated at line 193. “The reference MenW-CRM197 was also prepared and characterized. Differently from the multiple copies that were displayed on the nanoparticles under investigation, a final glycosylation degree (w/w) of 0.7 was determined, which resulted in an average of 5 chains of MenW oligosaccharides exposed on the protein.   

Additionally, Table 2 has been revised to include MenW-CRM197 details.

To satisfy the request to further discuss different factors that can influence the overall performance of glycoconjugates, we proposed the following modifications:

Line 23

The final sentence in the abstract “In particular, spherical shape induces a marked effect on saccharide antigen-specific immune response” was removed

Line 324

A final comment was introduced in the discussion: “Moreover, it is crucial to highlight that the nanoparticles being compared, although they share a similar molecular weight of the monomeric unit, contain distinct T-cell epitopes. This difference could potentially influence the immune responses they trigger, a topic that warrants further exploration”.

In light of these considerations, we propose to slightly modify the current title “Shape of protein nanoparticle impacts the immunogenicity of antimicrobial glycoconjugate vaccines” into : Impact of protein nanoparticle shape on the immunogenicity of antimicrobial glycoconjugate vaccines” that better reflects the concept that multiple mechanisms can potentially contribute to increase the performance of spherical nanoparticles.

Reviewer 2 Report

Comments and Suggestions for Authors

The work by Carboni et al studies the effect of the protein nanoparticle shape on the immunogenicity of glycoconjugate candidate vaccines. In particular, three different protein shapes conjugated with oligosaccharides from Neisseria meningitidis type W capsular polysaccharide where compared, namely the ring-shape and nanotubes of Pseudomonas  aeruginosa Hemolysin-corregulated protein 1 and the spherical Helicobacter Pylori ferritin. This latter resulted as the most effective among those tested. The work is well written and the results are of interest for the readers of the Journal. Only a few modifications are suggested.

In the introduction, it would beneficial to stress that part structures of microbial capsular polysaccharides have also been prepared via synthetic/chemo-enzymatic approaches (https://doi.org/10.1021/acscentsci.1c01479, https://doi.org/10.1039/C5OB00766F) and interesting results are obtained in this research area.

Also, line 49, “HBsAg antigen” seems not correct as the Ag stays for antigen.

Table 2, please correct “Sacchharide”.

Line 281, it should be “the highest” instead of “the higher”.

Line 297, “relying on”.

Lines 380-382; may the authors provide more details on the procedure followed? Is any additional solution added to wash out the unconjugated oligosaccharide?

Comments on the Quality of English Language

English level is satisfactory

Author Response

  • Reviewer 2:

Comments:

The work by Carboni et al studies the effect of the protein nanoparticle shape on the immunogenicity of glycoconjugate candidate vaccines. In particular, three different protein shapes conjugated with oligosaccharides from Neisseria meningitidis type W capsular polysaccharide where compared, namely the ring-shape and nanotubes of Pseudomonas  aeruginosa Hemolysin-co-egulated protein 1 and the spherical Helicobacter Pylori ferritin. This latter resulted as the most effective among those tested. The work is well written and the results are of interest for the readers of the Journal. Only a few modifications are suggested. English level is satisfactory.

Suggestions:

  1. In the introduction, it would beneficial to stress that part structures of microbial capsular polysaccharides have also been prepared via synthetic/chemo-enzymatic approaches (https://doi.org/10.1021/acscentsci.1c01479, https://doi.org/10.1039/C5OB00766F) and interesting results are obtained in this research area.

The requested modification to the introduction has been carried out, with the specified sentence being incorporated at line 36: “The field of glycoconjugate vaccines is continually advancing, improving our understanding of key carbohydrate characteristics that can influence the immunogenicity of glycoconjugates. This includes research on new and innovative techniques for creating saccharide antigens, such as synthetic or chemo-enzymatic methods, which have already yielded promising results (https://doi.org/10.1021/acscentsci.1c01479, https://doi.org/10.1039/C5OB00766F). Concurrently, there is a growing interest in identifying new carrier proteins that could potentially amplify vaccine efficacy. This is achieved by displaying multiple copies of an antigen, thereby mimicking its natural presentation by the pathogen”.

As per the recommendations of Reviewer 2, we have incorporated two additional references into the paper. Consequently, the numbering of the subsequent references has been updated to maintain the correct sequence.

  1. line 49, “HBsAg antigen” seems not correct as the Ag stays for antigen.

“HBsAg antigen” actually in line 53 was modified in “HBsAg” as requested.

5.Table 2, please correct “Sacchharide”.

“Sacchharide” in table 2 was modified in “Saccharide” as requested.

  1. Line 281, it should be “the highest” instead of “the higher”.

The word “highest” actually in line 290 was modified in “highest” as requested.

  1. Line 297, “relying on”.

“Relying into” actually in line 306 was modified in “relying on” as requested.

  1. Lines 380-382; may the authors provide more details on the procedure followed? Is any additional solution added to wash out the unconjugated oligosaccharide?

For enhanced clarity, the buffer employed for the washing process was incorporated into the M&M section. At line 393-396, the sentence was modified as follow: "The unconjugated oligosaccharide was extensively removed performing serial centrifugal filtration (30 kDa) with reducing buffer 500 mM NaCl, 50 mM TRIS pH 7.5, 10% glycerol with 5 mM DTT to maintain Hcp1cc in ring form”.
